# Carotid intima-media thickness in polycystic ovary syndrome and its association with hormone and lipid profiles

**Rhea Jabbour**[1]*, **Johannes Ott**[1], **Wolfgang Eppel**[2], **Peter Frigo**[1]

**1** Division of Gynecological Endocrinology and Reproductive Medicine, Department of Obstetrics and Gynecology, Medical University of Vienna, Vienna, Austria, **2** Division of Obstetrics and Feto-Maternal Medicine, Department of Obstetrics and Gynecology, Medical University of Vienna, Vienna, Austria

* rhea.jab@gmail.com

## Abstract

### Objective

Polycystic ovary syndrome (PCOS) has been associated with an increased risk of metabolic disturbances and cardiovascular disease. Intima-media thickness of the common carotid artery (CIMT) represents a valid surrogate marker of early systemic atherosclerosis. This study aimed to investigate if CIMT is increased in PCOS patients compared to healthy controls and if there is an association with hormone and metabolic profiles.

### Methods

In this prospective cross-sectional study, past medical history, anthropometrical measurements and hormonal, lipidemic and glycemic parameters were obtained in 41 PCOS patients and 43 age-matched healthy controls of similar body mass index (BMI) and frequency of smokers. B-mode ultrasound enabled CIMT measurement at the far wall of the left and right common carotid artery.

### Results

Patients with PCOS showed significantly increased CIMT values compared to healthy controls (0.49±0.04mm vs. 0.37±0.04mm respectively, $P$<0.001). They featured a generally increased cardiovascular risk profile. Correlation analysis showed a positive association between CIMT and the adverse metabolic risk profile. The diagnosis of PCOS was the strongest predictor of CIMT, even after multiple adjustments for BMI, age and smoking status ($\beta$ = 0.797, $P$<0.001, $R^2$ = 0.73). A model among oligomenorrhoic patients revealed a relationship between CIMT and the suspected duration of disease ($\beta$ = 0.373, $P$ = 0.021, $R^2$ = 0.14).

### Conclusions

PCOS patients are likely to feature signs of premature systemic atherosclerosis at a young age. Early exposure to adverse cardiovascular risk factors may possibly have long-term

**Data Availability Statement:** All relevant data are within the paper.

**Funding:** The authors received no specific funding for this work.

**Competing interests:** The authors have declared that no competing interests exist.

consequences on the vascular system. An early vessel screening might thus already be beneficial in these patients at a younger age.

## Introduction

Polycystic ovary syndrome (PCOS) represents one of the most common endocrinopathies in women of reproductive age, affecting approximately 4 to 7% of all women [1]. Clinical features include hyperandrogenism, ovulatory dysfunction and polycystic ovarian morphology on ultrasound (PCOM) [2,3]. The fundamental underlying defect in PCOS still remains unclear. However, PCOS seems to be a complex state of multifactorial origin, resulting from genetic components that interact with environmental factors. Intrinsic ovarian dysfunction, leading to dysregulated ovarian steroidogenesis with higher androgen production, neuroendocrine abnormalities and hyperinsulinemia are thought to play significant interactive roles [4,5]. In fact, PCOS seems to be closely associated with a metabolic disorder linked to insulin resistance (IR) [6], as 50–70% of PCOS patients show IR with compensatory hyperinsulinemia [7]. This state is characterized by IR in skeletal muscle, adipose tissue and liver, due to a post-binding defect in insulin receptor signaling caused by increased serine phosphorylation of the insulin receptor and the insulin receptor substrate-1 (IRS-1), but ovarian hypersensitivity to insulin [6,8]. Further clinical manifestations of PCOS include metabolic disturbances, such as abdominal obesity, dyslipidemia, impaired glucose tolerance, type II diabetes mellitus (DM), arterial hypertension and the metabolic syndrome [7,9,10]. IR being a key component of the pathophysiology of PCOS, treatment options with insulin-sensitizers, such as metformin or inositol isoforms [11], are increasingly being used due to their beneficial effects on metabolic and hormonal parameters.

The above-mentioned characteristics represent typical risk factors for early atherosclerosis [12], leading to an increased risk of cardiovascular disease [13]. In fact, a 2-fold increased likelihood of coronary heart disease or stroke [14] and higher prevalence of coronary artery and aortic calcification have been reported among PCOS patients [15]. Measurement of intima-media thickness of the common carotid artery (CIMT) using brightness mode ultrasound (B-mode) represents a valid surrogate marker of early systemic atherosclerosis [16,17,18]. This sensitive and noninvasive method thus enables screening for atherosclerosis and cardiovascular risk assessment. CIMT has been associated with an increased risk of coronary heart disease [19] and permits to predict the likelihood of future cardiovascular events, such as stroke and myocardial infarction [17,18,19].

There is evidence that CIMT is elevated in women suffering from PCOS [20,21], especially when they have passed forty [22,23]. However, CIMT increase can already be seen at a young age, such as in adolescence [24], although discrepant findings have been reported [21,25]. Previous studies have also shown an association between CIMT and lipid profile [26], as well as a relationship with hyperinsulinemia [26] and endogenous androgen levels [20], even though differing results have been reported. In fact, hyperandrogenism is said to lead to an increase in CIMT through its proatherogenic effect [20], but numerous studies have also demonstrated an inverse correlation of CIMT with androgen levels in women [27,28,29,30]. On the other hand, estrogen and sex hormone-binding globulin (SHBG) are found to have beneficial effects on lipid profile, thus leading to a reduced CIMT [29].

Despite increasing knowledge about PCOS in the past years, the findings about CIMT and its predictors in PCOS are still very controversial and little is known about the true impact of the disorder. The aim of the present study was therefore to investigate whether CIMT is

increased in PCOS patients compared to healthy control subjects, as well as to find out, if this cardiovascular risk factor is associated with hormone and metabolic profiles.

## Materials and methods

### Study participants

In this prospective cross-sectional study, 41 patients with PCOS between 18 and 34 years of age were consecutively recruited as they attended the endocrine outpatient department of Vienna General Hospital for medical consultation (Department of Gynecology, Division of Gynecological Endocrinology and Reproductive Medicine, Medical University of Vienna) between July 1st and September 30th 2015. Diagnosis of PCOS was made according to the Rotterdam ESHRE/ASRM criteria [3] if two out of the three following features were present: oligoovulation or anovulation, quantified by oligomenorrhea (≤6 menstrual periods in the past year [24]); clinical and/or biochemical signs of hyperandrogenism; PCOM, i.e. at least one ovary with presence of ≥12 follicles measuring 2–9 mm in diameter and/or increased ovarian volume (>10 ml) [3]. In this way, 4 different PCOS phenotypes were defined, i.e. phenotype 1 (hyperandrogenism, oligo-anovulation, PCOM), phenotype 2 (hyperandrogenism with oligo-anovulation), phenotype 3 (hyperandrogenism with PCOM) and phenotype 4 (oligo-anovulation with PCOM). Clinical hyperandrogenism included hirsutism, determined as Ferriman-Gallwey score (FGS) ≥8, and/or acne [24]. Biochemical hyperandrogenism was defined according to the local laboratory's normal reference ranges as at least one of the following conditions: dehydroepiandrosterone sulfate (DHEAS) >3.7 μg/mL, free testosterone >0.22 ng/mL, total testosterone >0.48 ng/mL, androstenedione >4.1 ng/mL. Hyperandrogenemia was considered mandatory in order to fulfill the criterion of hyperandrogenism in this study. Other etiologies of hyperandrogenism (e.g. congenital adrenal hyperplasia, androgen-secreting tumors, Cushing's syndrome, non-classic congenital adrenal hyperplasia using 17-hydroxy-progesterone (17-OHP) <2 ng/mL as surrogate parameter [3]) or secondary amenorrhea (anorexia, hyperprolactinemia, i.e. prolactin ≥31 ng/mL after polyethylene glycol immunoprecipitation [31]) were excluded. Thus, 26 newly diagnosed women with PCOS and 15 already diagnosed patients (1 month to 12 years ago) under no treatment for at least 3 months were included in this study.

43 age-matched, healthy and regularly menstruating female volunteers, drawn from the normal community and of similar body mass index (BMI) and frequency of smokers served as control subjects. PCOS was ruled out in all controls according to the Rotterdam ESHRE/ASRM criteria [3], as transvaginal ultrasound was performed between the third and the fifth day of their menstrual cycle in case of hyperandrogenism to exclude PCOS phenotype 3. Metabolic or cardiovascular diseases, e.g. diabetes mellitus, hyperlipidemia, arterial hypertension, coronary heart disease, thyroid dysfunction (i.e. thyroid-stimulating hormone (TSH) <0.44 μU/mL or ≥4.9 μU/mL) or BMI <17 kg/m$^2$ or >50 kg/m$^2$, lactation or pregnancy (beta-human chorionic gonadotropin (beta-hCG) levels >0.1 mU/mL), ongoing medication or hormonal contraception, current psychiatric disorders, heavy consumption of alcohol, caffeine or smoking (>20 cigarettes per day) were further ruled out in all subjects.

The study was approved by the Ethics Committee of the Medical University of Vienna (EK-Nr. 1197/2015). It was conducted in accordance with the Declaration of Helsinki. Written informed consent was obtained from all recruited subjects.

### Clinical and biochemical measurements

Initial physical examination included past medical history, age, BMI (BMI = weight$_{[kg]}$/height$_{[m]}{}^2$), waist-hip ratio (WHR = waist circumference (WC)$_{[cm]}$/hip circumference

(HC)$_{[cm]}$), FGS and acne. We further defined the suspected starting point of disease in PCOS patients as the onset of oligomenorrhea. Therefore, when oligomenorrhea started right at menarche, it was noted as starting 0 years after menarche, meaning that 0 years equals menarche. After a 5-minute-rest in a sitting position, resting heart rate, systolic (SBP) and diastolic (DBP) blood pressure were taken with a sphygmomanometer, SBP and DBP being defined by the means of two consecutive measurements on each arm. Pelvic ultrasound was further performed in PCOS patients and controls with hyperandrogenism.

Blood samples were obtained in the morning after overnight fasting in subjects with a normal carbohydrate diet, between the third and the fifth day of their menstrual cycle or randomly in PCOS cases with an irregular cycle for the measurement of hormonal, lipidemic and glycemic parameters, 25-hydroxy-vitamine D and C-reactive protein. Free androgen index (FAI) was calculated as follows: (total testosterone$_{[nmol/L]}$/SHBG$_{[nmol/L]}$)x100 [32]. Insulin sensitivity was estimated with homeostasis model assessment of insulin resistance (HOMA-IR = (glucose$_{[mmol/L]}$x insulin$_{[\mu U/mL]}$)/22.5) [7].

Biochemical and hormonal parameters were acquired by enzyme-linked immuno assays (ELISA). All analyses were performed at the routine hormone laboratory of Vienna General Hospital, the Department of Medical and Chemical Laboratory Diagnostics (http://www.kimcl.at).

## CIMT measurement

B-mode ultrasound images of the common carotid artery (CCA) were obtained within the next couple of days, using a 6- to 8-MHz high-resolution linear ultrasound probe for vascular/ small parts (GE Medical Systems Ultrasound LOGIQ 9), following a standardized protocol under optimal adjustment of depth, gain and focus and controlled temperature and light conditions. After a resting period of 10 minutes to enable heart rate and blood pressure stabilization, subjects were examined in supine position, with a 35-degree extension and a slight right- or left-turn of the head, depending on the explored side. Left and right CCA were scanned in different transversal and longitudinal planes by the same trained sonographer, who was sitting at the top end of the examination bed behind the patient [33]. Interference of CIMT estimation with a possible plaque or thrombus was thus excluded. After freezing a gray-scale image in a longitudinal plane, CIMT was measured directly during diastole at the far wall of both distal CCAs, as the distance between the lumen-intima and media-adventitia interfaces [16,20,33]. Three consecutive measurements of CIMT were conducted on each side at a distance of 5 millimeters, starting off 1 centimeter prior to the common carotid bulb [20,33] (Fig 1). This resulted in an average value for each left and right CCA. The outcome variable "CIMT" was defined within a single subject as the mean value calculated from left and right CIMT [25], the average from both sides being a more stable representation of CIMT [21]. The intraclass correlation coefficient, measured in 21 subjects, was 0.98 for the right and 0.99 for the left CIMT.

## Statistical analysis

Deviation of continuous data from a normal distribution was assessed through visual inspection of histograms and box and whisker plots. Continuous scaled variables were expressed as means ± standard deviations (SD) if normally distributed, or medians and interquartile ranges (IQR: Q1-Q3). Categorical variables were presented as numbers and percentages (%). Age matching was achieved by ensuring that overall age distribution in terms of mean and standard deviation in both groups is roughly the same.

Continuous parameters were compared between PCOS patients and controls using unpaired two-tailed Student's *t*-test when following a Gaussian distribution, or Mann-

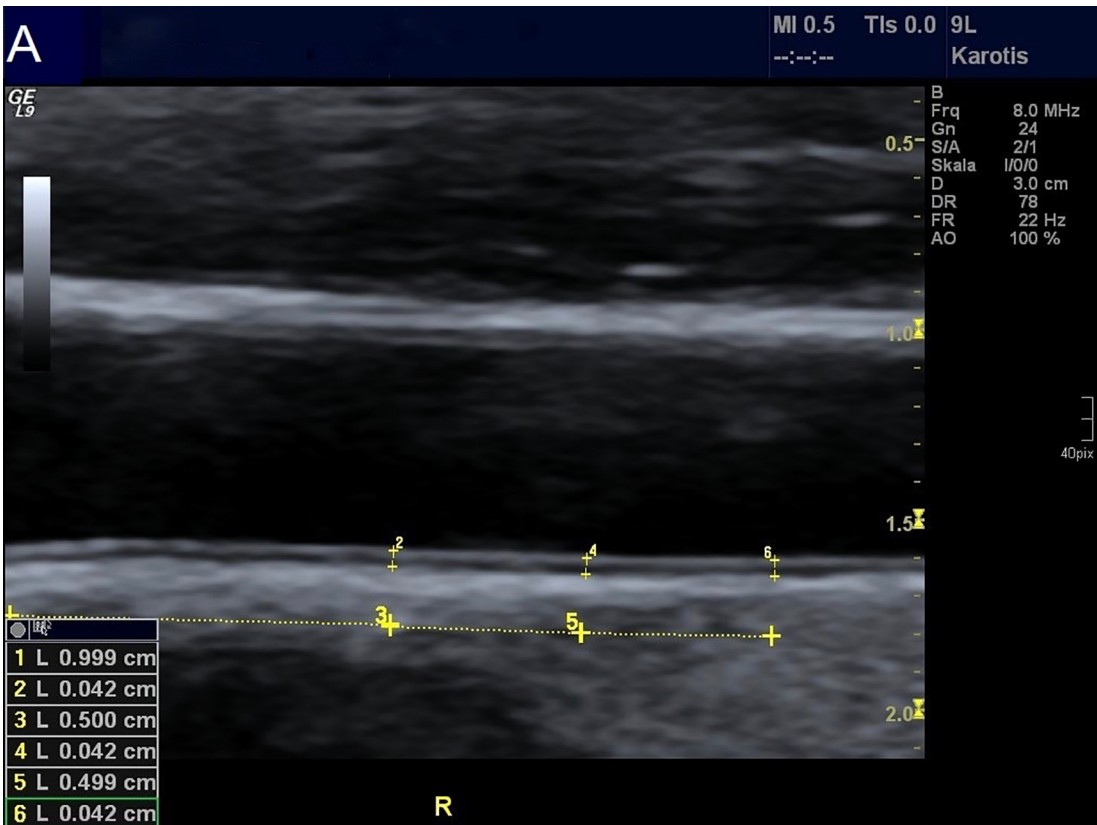

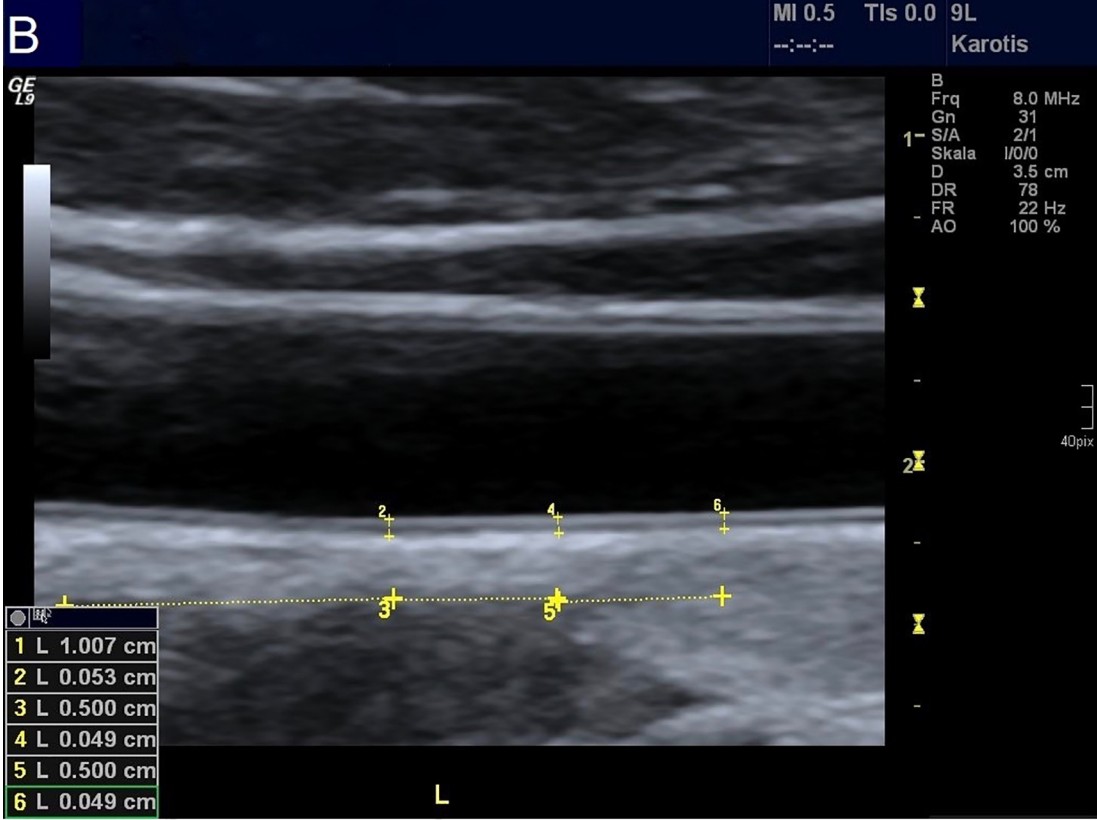

**Fig 1. CIMT measurement in PCOS patients and control subjects.** CIMT measurement at the far wall of the distal right (A) and left (B) common carotid artery as the distance between the lumen-intima and media-adventitia interfaces. (A) CIMT in a control subject. (B) CIMT in a PCOS patient.

Whitney $U$ test in case of skewed data. Nominal data were analyzed using $\chi^2$ test or Fisher's exact test. Associations between CIMT and different continuous variables were assessed by Spearman's rank correlation analysis (rho). The effect of nominal variables on CIMT was determined by Student's $t$-test.

Subsequently, multiple linear regression analysis was performed in order to identify independent factors that predict CIMT as the dependent continuous variable. PCOS status, BMI, age and smoking status were therefore entered sequentially as independent factors into the regression model, in order to assess the effect of PCOS and the potential confounding effect of the above-mentioned parameters. They were followed by factors found to be significantly associated with CIMT in individual bivariate correlation. First, the independent variable best correlated with CIMT was included, then the one with the next highest correlation, checking for multicollinearity and normal distribution of the residuals. Nominal variables, i.e. PCOS and smoking status, were defined as dummy variables, coded as 1 for "yes" and 0 for "no". Models were evaluated with the coefficient of determination $R^2$, which expresses the proportion of variability in the dependent variable, i.e. CIMT, explained by the model. Regression analysis was also done for the oligomenorrhoic PCOS group.

All data analyses were performed using the Statistical Package for the Social Sciences software (IBM SPSS Statistics) version 24.0. Two-sided $P$-values <0.05 were considered statistically significant. Sample size calculation was performed using "Java applets for power and sample size" (http://homepage.stat.uiowa.edu/~rlenth/Power/) with a significance α of 0.05 and a statistical power of 90% set to find a mean difference in CIMT of 0.08 mm [20,32].

## Results

### Subject characteristics

Baseline demographic and anthropometrical parameters and clinical characteristics of the study population are exposed in Table 1. Females with PCOS showed increased BMI ($P = 0.001$) and higher prevalence of smokers ($P = 0.007$), metabolic syndrome according to the NCEP ATP III criteria [7] ($P = 0.011$) and parental history of metabolic disorders (i.e. maternal PCOS or DM or paternal DM, $P = 0.001$) compared to controls, who were slightly older and had higher prevalence of alcohol consumption ($P = 0.001$). PCOS patients also exhibited preponderance of abdominal obesity, as revealed by significantly increased WC, WHR ($P<0.001$) and HC ($P = 0.019$) and elevated prevalence of WC>80 cm ($P = 0.001$). As expected [3], they further featured significantly higher prevalence of ovulatory dysfunction, hirsutism and hyperandrogenemia ($P<0.001$) and increased age at menarche ($P = 0.010$). Accordingly, 29 (70.7%) met the criteria of classic PCOS (phenotype 1), 3 (7.3%) exhibited ovulatory PCOS (phenotype 3) and 9 (22%) had non-hyperandrogenic PCOS (phenotype 4). None showed phenotype 2. In oligomenorrhoic patients (phenotypes 1 and 4), onset of oligomenorrhea had mostly occurred at menarche, resulting in a suspected duration of disease since onset of 9.1 ± 4.8 years.

### Carotid intima-media thickness

CIMT values were significantly higher in PCOS patients than controls (0.49 ± 0.04 mm vs. 0.37 ± 0.04 mm respectively, $P<0.001$, Table 2), resulting in a mean difference of 0.12 mm

**Table 1. Baseline demographic and anthropometrical parameters and clinical and biochemical diagnostic criteria of the polycystic ovary syndrome among the study population.**

| Variables | PCOS (n = 41) | Controls (n = 43) | P-value |
|---|---|---|---|
| Age (years) | 24 ± 4 | 25 ± 4 | 0.296[a] |
| Age at menarche (years) | 14 (12–14)[i] | 12 (12–13)[i] | **0.010[b]** |
| Single | 16 (39.0) | 24 (55.8) | 0.124[c] |
| Nulligravida | 33 (80.5) | 38 (88.4) | 0.318[c] |
| BMI (kg/m$^2$) | 26.33 ± 7.30 | 21.91 ± 3.22 | **0.001[a]** |
| WC (cm) | 91.2 ± 18.0 | 78.4 ± 9.8 | **< 0.001[a]** |
| WC > 80cm | 27 (65.9) | 13 (30.2) | **0.001[c]** |
| HC (cm) | 104 ± 14 | 98 ± 9 | **0.019[a]** |
| WHR | 0.87 ± 0.07 | 0.80 ± 0.06 | **< 0.001[a]** |
| Metabolic syndrome (NCEP)[ii] | 6 (14.6) | 0 | **0.011[d]** |
| Current smokers | 15 (36.6) | 5 (11.6) | **0.007[c]** |
| Alcohol consumption | 16 (39.0) | 32 (74.4) | **0.001[c]** |
| Caffeine consumption | 32 (78.0) | 33 (76.7) | 0.886[c] |
| Parental history of metabolic disorders[iii] | 17 (41.5) | 4 (9.3) | **0.001[c]** |
| Oligomenorrhea/amenorrhea | 38 (92.7) | 0 | **< 0.001[c]** |
| Menstrual periods per year | 4 (2–6)[i] | 12±1 | **< 0.001[b]** |
| Hirsutism (FGS ≥ 8) | 29 (70.7) | 8 (18.6) | **< 0.001[c]** |
| FGS | 11 (7–23)[i] | 4 (2–7)[i] | **< 0.001[b]** |
| Acne | 32 (78.0) | 21 (48.8) | **0.006[c]** |
| Hyperandrogenemia[iv] | 32 (78.0) | 6 (14.0) | **< 0.001[c]** |
| *DHEAS > 3.7μg/mL | 19 (46.3) | 4 (9.3) | **< 0.001[c]** |
| *fT > 0.22 ng/mL | 17 (41.5) | 0 | **< 0.001[c]** |
| *TT > 0.48 ng/mL | 28 (68.3) | 3 (7.0) | **< 0.001[c]** |
| *A > 4.1 ng/mL | 14 (35.0) | 0 | **< 0.001[c]** |
| PCOM | 41 (100) | - | - |
| Oligomenorrhea onset (years after menarche)[v] | 0.0 (0.0–3.0)[i] | - | - |
| Suspected duration of disease since onset (years)[v] | 9.1 ± 4.8 | - | - |

Values are presented as means ± standard deviations, numbers and percentages (%) or [i]medians and interquartile ranges (IQR).

P-values < 0.05 were considered statistically significant and marked in bold.

[a]Student's t-test

[b]Mann-Whitney U test

[c]$\chi^2$ test

[d]Fisher's exact test.

[ii]Metabolic syndrome according to the NCEP ATP III criteria [7].

[iii]Mother with polycystic ovary syndrome or diabetes mellitus or father with diabetes mellitus.

[iv]Defined as the increase of at least one androgen (DHEAS, fT, TT, A).

[v]In PCOS patients with phenotypes 1 and 4 (n = 38).

BMI: body mass index; WC: waist circumference; HC: hip circumference; WHR: waist-hip ratio; FGS: Ferriman-Gallwey score; DHEAS: dehydroepiandrosterone sulfate; fT: free testosterone; TT: total testosterone; A: androstenedione; PCOM: polycystic ovarian morphology on ultrasound.

between both study groups. They ranged from 0.40 to 0.63 mm in women with PCOS (n = 41) and from 0.31 to 0.46 mm in control subjects (n = 43, Fig 2). Key exploratory endpoints, i.e. right and left CIMT and maximal and minimal CIMT, were also significantly higher in PCOS patients (P<0.001).

**Table 2. Cardiovascular parameters.**

| Variables | PCOS (n = 41) | Controls (n = 43) | P-value |
|---|---|---|---|
| CIMT (mm) | 0.49 ± 0.04 | 0.37 ± 0.04 | < **0.001**[a] |
| Right CIMT (mm) | 0.49 ± 0.05 | 0.37 ± 0.04 | < **0.001**[a] |
| Left CIMT (mm) | 0.50 ± 0.05 | 0.37 ± 0.04 | < **0.001**[a] |
| CIMTmax (mm) | 0.56 ± 0.06 | 0.41 ± 0.04 | < **0.001**[a] |
| CIMTmin (mm) | 0.42 ± 0.04 | 0.33 ± 0.04 | < **0.001**[a] |
| SBP (mmHg) | 113 ± 13 | 109 ± 10 | 0.088[a] |
| DBP (mmHg) | 72 ± 9 | 71 ± 8 | 0.533[a] |
| Heart rate (bpm) | 70 ± 11 | 71 ± 10 | 0.484[a] |
| hs-CRP (mg/dL) | 0.07 (0.05–0.25) | 0.07 (0.04–0.17) | 0.343[b] |

Values are presented as means ± standard deviations or medians and interquartile ranges (IQR).

P-values < 0.05 were considered statistically significant and marked in bold.

[a]Student's t-test

[b]Mann-Whitney U test.

CIMT: carotid intima-media thickness; SBP: systolic blood pressure; DBP: diastolic blood pressure; hs-CRP: high-sensitivity C-reactive protein.

## Cardiovascular, metabolic and hormone profiles

Cardiovascular, metabolic and hormonal parameters are provided in Tables 2 and 3. In comparison to controls, PCOS patients showed higher levels of low-density lipoprotein cholesterol (LDL-C), total cholesterol/high-density lipoprotein cholesterol ratio (TC/HDL-C ratio) and apolipoprotein B ($P<0.001$), triglycerides ($P = 0.001$) and TC ($P = 0.005$), as well as lower levels of HDL-C ($P = 0.044$), revealing an adverse lipid profile. Increased levels of C-peptide and fasting insulin ($P<0.05$) also indicated higher endogenous insulin secretion. PCOS patients further presented with significantly higher levels of all measured androgens, FAI, 17-OHP, luteinizing hormone (LH), LH/FSH (follicle-stimulating hormone) ratio and Anti-Müllerian hormone (AMH) ($P\leq0.001$), and lower levels of SHBG ($P = 0.006$) and FSH ($P = 0.008$).

## Correlation and regression analysis

Several cardiovascular risk factors and hormonal parameters were found to be significantly positively correlated with CIMT in our subjects (Table 4), including total testosterone, free testosterone, androstenedione, FAI, AMH, LH/FSH ratio, WC, WHR, BMI, FGS and apolipoprotein B ($P<0.001$), as well as 17-OHP, LDL-C, smoking, TC, triglycerides, TC/HDL-C ratio, DHEAS and SBP ($P<0.05$). Besides, SHBG showed a significant negative relationship with CIMT ($P = 0.023$). When carried out separately in PCOS patients, correlation analysis revealed a significant positive association of CIMT with the suspected duration of disease ($P = 0.012$). Additionally, unpaired two-tailed Student's t-test showed significantly higher CIMT values among the entire study population in case of hyperandrogenemia (0.47 ± 0.06 (n = 38) vs. 0.40 ± 0.07 (n = 46), $P<0.001$) and parental history of metabolic disorders (0.48 ± 0.05 (n = 21) vs. 0.41 ± 0.07 (n = 63), $P<0.001$).

Multiple linear regression analysis was carried out in order to identify independent factors that predict CIMT as the dependent continuous variable (Table 5). When considering PCOS patients and controls as a group, the diagnosis of PCOS was the strongest predictor of CIMT (β = 0.836, $P<0.001$), explaining 70% of its variability ($P<0.001$, model 1). In order to assess the potential confounding effect of obesity, age and smoking on CIMT, BMI, age and smoking

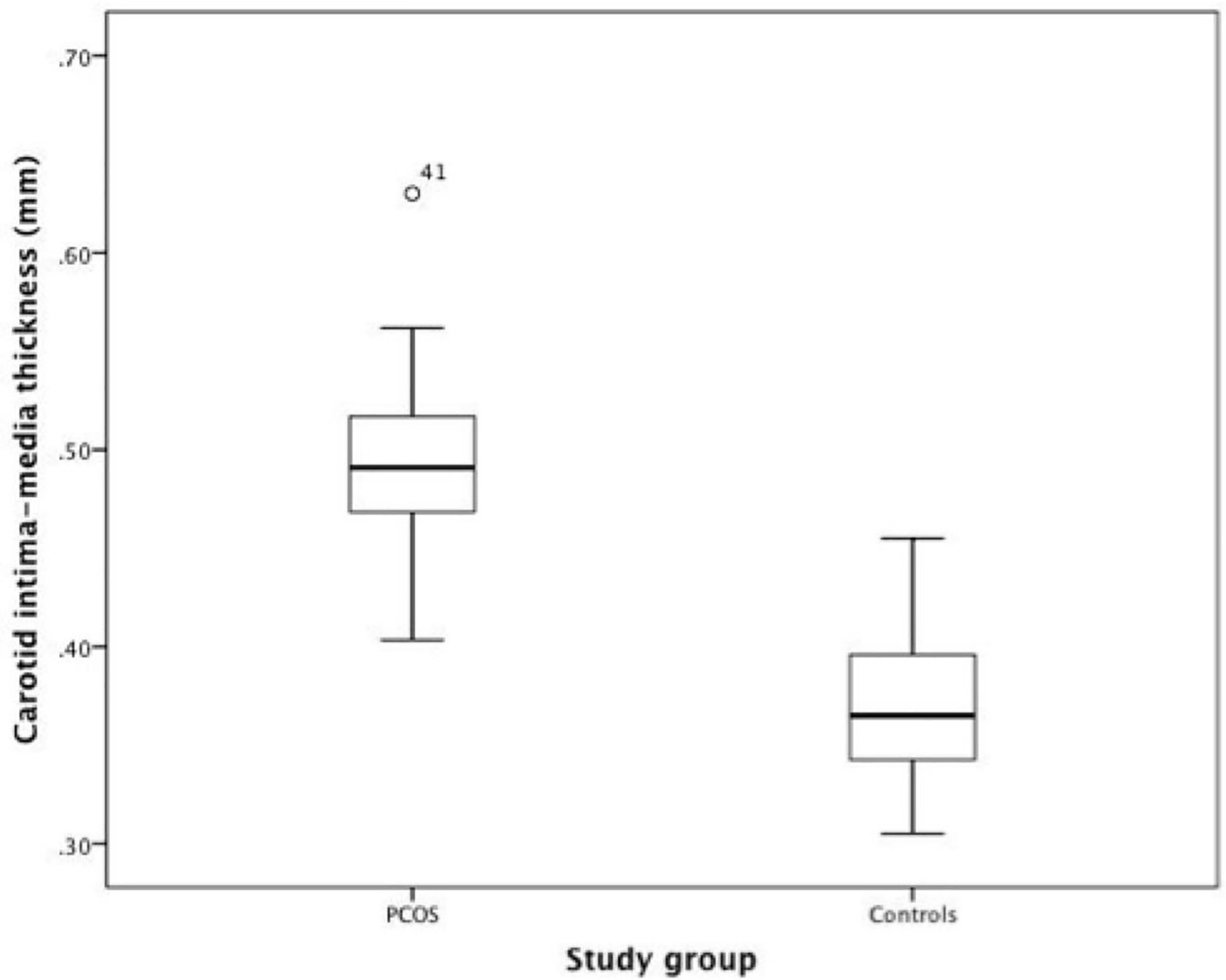

**Fig 2. Box and whisker plots of CIMT.** Among each group of interest (i.e. PCOS and controls). Box and whisker plots enable to summarize descriptive analysis of CIMT. They comprise the median (bold horizontal line) and the interquartile range (IQR), represented by a box, the first and third quartile being the borders of this box. Whiskers show values that are within 1.5 IQRs (vertical lines extending from the box). The small circle indicates an outlier (over 1.5 IQRs).

status were entered sequentially as independent factors into the regression model. Only BMI represented an independent positive predictor of CIMT ($\beta$ = 0.145, $P$ = 0.025) among these parameters, raising the predicted variability in CIMT to 72% ($P<0.001$, model 2). However, PCOS status remained the primary predictor of CIMT, even after multiple adjustments for BMI, age and smoking status ($P<0.001$, $R^2$ = 0.73, models 3 and 4), therefore indicating an independent effect of PCOS on CIMT ($\beta$ = 0.797, $P<0.001$). No statistical significance was found for further hormonal and cardiovascular risk factors, probably due to multicollinearity with the PCOS status. When performed separately in oligomenorrhoic PCOS patients (n = 38), regression analysis revealed suspected duration of disease as an independent positive predictor of CIMT ($\beta$ = 0.373, $P$ = 0.021), accounting for 14% of the variability in CIMT ($P$ = 0.021, model 5).

**Table 3. Metabolic parameters and hormone profile.**

| Variables | PCOS (n = 41) | Controls (n = 43) | P-value |
|---|---|---|---|
| **Metabolic parameters** | | | |
| Triglycerides (mg/dL) | 81 ± 48 | 54 ± 17 | **0.001**[a] |
| Total cholesterol (mg/dL) | 175 ± 27 | 175 ± 27 | **0.005**[a] |
| LDL-cholesterol (mg/dL) | 98.6 ± 24.4 | 79.5 ± 17.5 | **< 0.001**[a] |
| HDL-cholesterol (mg/dL) | 60 (45–73) | 68 (58–77) | **0.044**[b] |
| TC/HDL-C ratio | 3.19 ± 1.13 | 2.39 ± 0.48 | **< 0.001**[a] |
| TC/HDL-C ratio > 4 | 9 (22.0)[i] | 0[i] | **0.001**[c] |
| Apolipoprotein B (mg/dL) | 85 ± 20 | 70 ± 11 | **< 0.001**[a] |
| Apolipoprotein A1 (mg/dL) | 147 ± 30 | 157 ± 27 | 0.093[a] |
| Lipoprotein(a) (mg/dL) | 14 (7–58) | 11 (7–33) | 0.644[b] |
| Fasting glucose (mg/dL) | 84 ± 6 | 82 ± 6 | 0.338[a] |
| HbA1c (%) | 5.1 ± 0.3 | 5.1 ± 0.2 | 0.931[a] |
| C-peptide (ng/mL) | 2.3 ± 1.1 | 1.6 ± 0.5 | **0.001**[a] |
| Fasting insulin (µU/mL) | 7.8 (5.5–16.0) | 6.3 (4.8–9.2) | **0.047**[b] |
| HOMA-IR | 1.56 (1.06–3.24) | 1.38 (0.94–1.84) | 0.051[b] |
| 25-OH-Vit D (nmol/L) | 57.9 (29.1–80.9) | 63.6 (51.5–77.5) | 0.080[b] |
| **Hormone profile** | | | |
| Total testosterone (ng/mL) | 0.56 ± 0.20 | 0.29 ± 0.15 | **< 0.001**[a] |
| Free testosterone (ng/mL) | 0.21 ± 0.13 | 0.08 ± 0.04 | **< 0.001**[a] |
| Androstenedione (ng/mL) | 3.50 ± 1.29 | 2.04 ± 0.82 | **< 0.001**[a] |
| DHEAS (µg/mL) | 3.40 ± 1.29 | 2.55 ± 1.03 | **0.001**[a] |
| SHBG (nmol/L) | 53.8 ± 31.2 | 73.3 ± 32.0 | **0.006**[a] |
| FAI | 3.90 (2.26–6.34) | 1.61 (0.81–2.22) | **< 0.001**[b] |
| 17-OHP (ng/mL) | 1.00 ± 0.56 | 0.61 ± 0.37 | **< 0.001**[a] |
| LH (mU/mL) | 12.9 ± 7.6 | 6.3 ± 2.2 | **< 0.001**[a] |
| FSH (mU/mL) | 5.6 ± 1.6 | 6.8 ± 2.4 | **0.008**[a] |
| LH/FSH | 2.39 ± 1.38 | 1.01 ± 0.45 | **< 0.001**[a] |
| AMH (ng/mL) | 12.30 (10.20–17.70) | 5.81 (2.55–9.11) | **< 0.001**[b] |
| Prolactin (ng/mL) | 12.5 ± 7.1 | 13.7 ± 5.3 | 0.368[a] |
| TSH (µU/mL) | 1.78 ± 0.85 | 1.96 ± 0.92 | 0.347[a] |
| Free T4 (ng/dL) | 1.25 ± 0.19 | 1.24 ± 0.17 | 0.798[a] |

Values are presented as means ± standard deviations (SD), medians and interquartile ranges (IQR) or [i]numbers and percentages (%).

P-values < 0.05 were considered statistically significant and marked in bold.

[a]Student's t-test

[b]Mann-Whitney U test

[c]Fisher's exact test.

LDL-cholesterol: low-density lipoprotein cholesterol; HDL-cholesterol: high-density lipoprotein cholesterol; TC/HDL-C ratio: total cholesterol/HDL-cholesterol ratio; HbA1c: glycosylated hemoglobin; HOMA-IR: homeostasis model assessment of insulin resistance; 25-OH-Vit D: 25-hydroxy-vitamine D; DHEAS: dehydroepiandrosterone sulfate; SHBG: sex hormone-binding globulin; FAI: free androgen index; 17-OHP: 17-hydroxyprogesterone; LH: luteinizing hormone; FSH: follicle-stimulating hormone; AMH: anti-Müllerian hormone; TSH: thyroid-stimulating hormone; T4: thyroxine.

## Discussion

The present study aimed to assess evidence of early systemic atherosclerosis in PCOS with measurement of CIMT and its possible association with hormone and metabolic profiles. Our results revealed significantly increased CIMT in PCOS patients when compared to healthy controls. This increase was further found to be independent of BMI, age and smoking status,

**Table 4. Correlation analysis with CIMT and baseline demographic characteristics, metabolic and hormone profiles with Spearman's correlation coefficient (rho).**

| Variables | Rho | *P*-value |
|---|---|---|
| **In all study subjects (n = 84)**p: | | |
| Total testosterone | 0.528 | **< 0.001** |
| Free testosterone | 0.505 | **< 0.001** |
| Androstenedione | 0.493 | **< 0.001** |
| FAI | 0.466 | **< 0.001** |
| DHEAS | 0.241 | **0.027** |
| SHBG | -0.247 | **0.023** |
| 17-OHP | 0.357 | **0.001** |
| LH/FSH | 0.436 | **< 0.001** |
| AMH | 0.458 | **< 0.001** |
| Triglycerides | 0.291 | **0.007** |
| Total cholesterol | 0.301 | **0.005** |
| LDL-cholesterol | 0.345 | **0.001** |
| HDL-cholesterol | -0.112 | 0.310 |
| TC/HDL-C ratio | 0.286 | **0.008** |
| Apolipoprotein B | 0.375 | **< 0.001** |
| Fasting insulin | 0.178 | 0.105 |
| Age | 0.053 | 0.630 |
| BMI | 0.398 | **< 0.001** |
| Waist circumference | 0.425 | **< 0.001** |
| Waist-hip ratio | 0.420 | **< 0.001** |
| SBP | 0.222 | **0.043** |
| Ferriman-Gallwey score | 0.394 | **< 0.001** |
| Smoking (py) | 0.306 | **0.005** |
| **In oligomenorrhoic PCOS subjects[a] (n = 38):** | | |
| Suspected duration of disease | 0.403 | **0.012** |

*P*-values < 0.05 were considered statistically significant and marked in bold.

[a]PCOS patients with phenotypes 1 and 4.

FAI: free androgen index; DHEAS: dehydroepiandrosterone sulfate; SHBG: sex hormone-binding globulin; 17-OHP: 17-hydroxyprogesterone; LH/FSH: luteinizing hormone/follicle-stimulating hormone; AMH: anti-Müllerian hormone; LDL-cholesterol: low-density lipoprotein cholesterol; HDL-cholesterol: high-density lipoprotein cholesterol; TC/HDL-C ratio: total cholesterol/HDL-cholesterol ratio; BMI: body mass index; SBP: systolic blood pressure; Py: pack years.

given that PCOS status remained the primary predictor of CIMT after covariate adjustment. These findings suggest that the disorder itself is playing a causative role in CIMT increase.

As described in former studies, measurement of CIMT using B-mode ultrasound represents a valid surrogate marker of subclinical systemic atherosclerosis and enables cardiovascular risk stratification [17,18]. There is evidence that women with PCOS feature an increased CIMT compared to healthy controls [20,21,34], with reported mean differences in CIMT ranging from 0.06 mm [35] to 0.14 mm [24,36]. The mean difference of 0.12 mm found in our study is therefore in accordance with literature, and so are our measured CIMT values among each group of interest, given that a meta-analysis [21] indicates a mean CIMT ranging from 0.41 to 0.75 mm in PCOS and from 0.33 to 0.74 mm in controls in previous studies. Differences in the prevalence of PCOS phenotypes and thus cardiovascular risk profiles between studies can

**Table 5. Multiple linear regression models of CIMT as dependent variable.**

| Predictors | B[a] | SE[b] | β[c] | P-value |
|---|---|---|---|---|
| **In all study subjects (n = 84):** | | | | |
| Model 1 (R² = 0.70)[d] | | | | |
| PCOS | 0.122 | 0.009 | 0.836 | **< 0.001** |
| Constant | 0.371 | 0.006 | | **< 0.001** |
| Model 2 (R² = 0.72)[d] | | | | |
| PCOS | 0.114 | 0.009 | 0.783 | **< 0.001** |
| BMI | 0.002 | 0.001 | 0.145 | **0.025** |
| Constant | 0.332 | 0.018 | | **< 0.001** |
| Model 3 (R² = 0.73)[d] | | | | |
| PCOS | 0.115 | 0.009 | 0.793 | **< 0.001** |
| BMI | 0.002 | 0.001 | 0.152 | **0.018** |
| Age | 0.002 | 0.001 | 0.111 | 0.061 |
| Constant | 0.281 | 0.032 | | **< 0.001** |
| Model 4 (R² = 0.73)[d] | | | | |
| PCOS | 0.116 | 0.009 | 0.797 | **< 0.001** |
| BMI | 0.002 | 0.001 | 0.159 | **0.018** |
| Age | 0.002 | 0.001 | | 0.065 |
| Smoking | -0.004 | 0.011 | 0.110 | 0.705 |
| Constant | 0.280 | 0.032 | -0.024 | **< 0.001** |
| **In oligomenorrhoic PCOS subjects[i] (n = 38):** | | | | |
| Model 5 (R² = 0.14)[e] | | | | |
| Suspected duration of disease | 0.003 | 0.001 | 0.373 | **0.021** |
| Constant | 0.460 | 0.015 | | **< 0.001** |

P-values < 0.05 were considered statistically significant and marked in bold.

[a]Unstandardized coefficients for determining the regression equation

[b]Standard error

[c]Standardized coefficient.

P-value of the model:

[d]< 0.001

[e]0.021.

[i]PCOS patients with phenotypes 1 and 4.

BMI: Body mass index.

explain discrepant findings regarding elevated CIMT in literature [21,25]. In fact, the preva-
lence of total and abdominal obesity, metabolic disturbances and cardiovascular risk factors is
found to decrease from phenotype 1 to 4, just like the severity of hyperandrogenism and men-
strual disturbances [3,10,37,38].

In the present study, several cardiovascular risk factors and hormonal parameters were
found to be significantly associated with CIMT, i.e. visceral obesity, dyslipidemia, hyperandro-
genemia, AMH, smoking, SBP and parental history of metabolic disorders.

Obesity is a common finding in PCOS and has been linked to increased CIMT in young
and middle-aged women [39]. As a matter of fact, women with PCOS are found to exhibit cen-
tral body fat distribution regardless of BMI, related to hyperandrogenemia and hyperinsuline-
mia [40]. However, adipose tissue is metabolically active and increased visceral fat thus
enhances secretion of free fatty acids (FFA), adipokines, i.e. adiponectin and leptin, growth
factors and inflammatory cytokines, e.g. TNF-α [41]. Those substances have a crucial impact
on metabolism and cardiovascular system. Visceral adipose tissue is therefore a source of

inflammation, the latter representing a key component of atherosclerosis development [42]. Markers of premature atherosclerosis, such as C-reactive protein (CRP), homocysteine and endothelial dysfunction were shown to be involved in CIMT increase [27,32,36]. Visceral obesity is further associated with dyslipidemia and IR, consequently leading to increased risk of DM and cardiovascular disease [40,43,44]. A positive relationship has thus been described with early signs of vascular damage and CIMT increase [32,45]. The higher prevalence of WC >80 cm in our PCOS patients, as defined in European women by the WHO [46], therefore indicates an increased risk of metabolic and cardiovascular complications, WC being a better predictor than WHR or BMI [43]. Metabolic syndrome according to the NCEP ATP III [7] being also more prevalent in our patients and former studies [24] further contributes to dyslipidemia and IR, leading to enhanced atherosclerosis and eventually CIMT increase.

Consistent with literature, PCOS patients featured an adverse lipid profile, including a higher TC/HDL-C ratio, which represents a risk factor for coronary heart disease [24,47]. Dyslipidemia is believed to be the consequence of increased visceral fat [32] and androgen levels [48,49]. Previous studies further confirmed the association found between CIMT and an adverse lipid profile, CIMT being positively associated with the atherogenic LDL-C, TC and triglycerides and negatively correlated with the antiatherogenic HDL-C [26]. As a result, women with PCOS are at increased risk of early systemic atherosclerosis [47].

The positive association of androgen levels with CIMT and higher CIMT values found in case of hyperandrogenemia are in accordance with previous findings [20], suggesting that CIMT increase in PCOS is being caused by hyperandrogenism. High levels of androgens are believed to lead to an increased CIMT through a direct effect on the vascular system. In fact, testosterone has proatherogenic effects on macrophage function by facilitating the uptake of modified lipoproteins [50] and enhancing monocyte adhesion to vascular endothelial cells [51]. Furthermore, androgens regulate lipolysis and lipogenesis in women, having an impact on lipid metabolism and adipocyte production [52]. Testosterone has been found to inhibit catecholamine-stimulated lipolysis in subcutaneous fat cells through decreased expression of hormone-sensitive lipase (HSL) and β2-adrenoceptors [53]. However, it seems to exert the opposite effect on visceral fat, leading to an increased sensitivity to catecholamines through upregulation of lipolytic β3-adrenergic receptors and HSL [49,54] and a reduced sensitivity to the antilipolytic effect of insulin [40,55]. As a consequence, the elevated release of FFA affects liver functioning and insulin signaling, leading to dyslipidemia, hyperinsulinemia, glucose intolerance and IR. It also reduces hepatic SHBG production, amplifying hyperandrogenemia [56]. Androgens further lead to an adverse lipid profile through inhibition of lipoprotein lipase (LPL) in adipose tissue [48,49]. Moreover, serum testosterone has been associated with advanced glycation end products (AGEs) [57]. These proinflammatory and oxidant mediators result from non-enzymatic glycation of proteins or lipids and are believed to play a contributive role in PCOS development [58,59], given that elevated levels have been reported [10,38,57,58]. AGEs further exert a proatherogenic effect on the vascular system [60]. They have been related to tissue damaging in the framework of atherosclerosis and may therefore contribute to the elevated cardiovascular risk in PCOS [38,57]. Besides, the role of testosterone as crucial determinant of CIMT increase is further emphasized by the positive relationship of CIMT seen with FAI, which represents a good reflection of testosterone action. Furthermore, the observed association between CIMT and hirsutism demonstrates the impact played by dihydrotestosterone (DHT), which leads to similar regional differences concerning lipolysis as testosterone [48,53]. The positive relationship of CIMT found with AMH further underlines the impact of hyperandrogenemia. In fact, granulosa cells of growing ovarian follicles in PCOS secrete elevated levels of AMH, which inhibit aromatase activity, thus contributing to androgen excess [61].

Smoking and SBP were also positively associated with CIMT. Indeed, smoking enhances the progression of extracoronary atherosclerosis [12] and hypertension is involved in the pathogenesis of vascular damage and thus atherosclerosis [62]. Moreover, higher CIMT values were seen in case of parental history of metabolic disorders, putting an emphasis on genetic predisposition in PCOS. Despite showing no correlation with CIMT in our study, higher endogenous insulin secretion, as found in our patients, is thought to contribute to CIMT increase. In fact, insulin exerts an atherogenic effect by increasing cholesterol transport into arteriolar smooth muscle cells. In this way, it stimulates their proliferation and endogenous cholesterol and collagen synthesis, eventually leading to elevated formation of lipid plaques [63].

Finally, our results from multiple regression analysis are in accordance with previous studies, in which the diagnosis of PCOS and BMI were found to predict CIMT [24,27], with PCOS status being the strongest predictor, independent of BMI [27]. Furthermore, our study revealed suspected duration of disease as predictor of CIMT, which is a new finding that has not been reported so far to our knowledge. Therefore, oligomenorrhoic patients seem to feature higher CIMT the earlier onset of disease occurs. Although suspected duration of disease is related to the subjects' age, it also depends on the age of menarche that varies in each patient. Besides, PCOS patients have later menarche than healthy controls [30,64,65], therefore it does not totally equate to the age. The observed relationship between suspected duration of disease and CIMT in oligomenorrhoic patients thus suggests even more that PCOS itself contributes to the enhancement of atherosclerosis, probably due to hyperandrogenism, an adverse lipid profile and hyperinsulinemia.

However, although women with PCOS feature increased CIMT and metabolic and cardiovascular risk at a younger age, there is evidence suggesting that they do not show higher prevalence of cardiovascular events than controls after menopause [30,64,65]. Possible explanations could be a protective effect of delayed menopause with a consequently prolonged estrogen exposure in PCOS [30,64,65] or even hyperandrogenism itself [64,65] in peri- and postmenopausal PCOS patients, as indicated by previous studies [27,28,29], probably mainly due to enzymatic conversion to estrogen. Moreover, differences in cardiovascular risk factors (e.g. diabetes, abdominal obesity) between PCOS and controls seem to be less preponderant in aging women, explaining the similar cardiovascular morbidity and mortality later in life [65].

Strengths of this study include the comparable sample size and age in both groups of interest. Controls were slightly older than PCOS patients, enabling to exclude even more a potential confounding effect of age on CIMT. All subjects further benefited from a thorough hormone, metabolic and cardiovascular assessment. Regarding limitations of the current study, even though most patients featured the classic PCOS phenotype, 29.3% belonged to phenotypes 3 or 4, with lesser severity of clinical and metabolic manifestations [3]. This could explain the only mild metabolic disturbances seen in our patients. Given that 9 PCOS patients without hyperandrogenism (i.e. phenotype 4) and 6 controls with hyperandrogenemia were included in the study, the impact of androgen excess on CIMT may have been underestimated and thus no significant association was found in multiple regression analysis. Besides, the presence of acne was quantified subjectively, thus no major focus should be put on its prevalence. Moreover, estrogen and progesterone effect on CIMT could not be examined in our study, given that blood tests were not always performed during follicular phase in oligomenorrhoic PCOS patients. Analyses should therefore be conducted in these women after progestin-induced withdrawal bleeding. Further studies with a larger sample size, especially among each PCOS phenotype, are required in order to unravel the exact underlying pathogenetic mechanisms in PCOS and to identify which components of this disorder have the greatest effect on CIMT.

## Conclusions

The present study indicates that patients with PCOS are likely to feature signs of premature systemic atherosclerosis already at a young age, as shown by their elevated CIMT values. Our findings further suggest that early exposure to adverse cardiovascular risk factors in the framework of this disorder may possibly have long-term consequences on the vascular system, given that the duration of disease seems to have a predictive impact on the extent of CIMT increase. In fact, metabolic disturbances, including visceral obesity, IR with compensatory hyperinsulinemia and dyslipidemia were found to be involved. Moreover, our results revealed that hyperandrogenism, a central feature of PCOS, represents a crucial determinant of CIMT elevation in women with this disorder and is thus implicated in the enhancement of atherosclerosis. However, there seems to exist a still unexplained independent effect of PCOS. A possible contributing factor to the elevated cardiovascular risk could be the proatherogenic role of AGEs, leading to tissue damage in the framework of atherosclerosis. PCOS patients thus seem to be at greater risk for early systemic atherosclerosis. This emphasizes the importance of thorough metabolic and cardiovascular evaluation in all women with PCOS. An early vessel screening might therefore already be beneficial in these patients at a younger age.

## Acknowledgments

Thanks to the Departments of Radiology and Neurology of the Medical University of Vienna for their support. We also thank Georg Dorffner, Ph.D., M.S. (Center for Medical Statistics, Informatics and Intelligent Systems, Institute of Artificial Intelligence) and Christian Göbl, M. D. (Department of Obstetrics and Gynecology, Medical University of Vienna) for their statistical advice.

## Author Contributions

**Conceptualization:** Rhea Jabbour, Peter Frigo.

**Data curation:** Rhea Jabbour, Peter Frigo.

**Formal analysis:** Rhea Jabbour.

**Investigation:** Rhea Jabbour, Peter Frigo.

**Methodology:** Rhea Jabbour, Peter Frigo.

**Project administration:** Rhea Jabbour.

**Resources:** Rhea Jabbour, Peter Frigo.

**Supervision:** Peter Frigo.

**Validation:** Rhea Jabbour, Peter Frigo.

**Visualization:** Rhea Jabbour, Johannes Ott, Peter Frigo.

**Writing – original draft:** Rhea Jabbour.

**Writing – review & editing:** Rhea Jabbour, Johannes Ott, Wolfgang Eppel, Peter Frigo.

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
