## [Decision Letter · Decision Letter 0]

18 Dec 2019

PONE-D-19-30548

Carotid intima-media thickness in polycystic ovary syndrome and its association with hormone and lipid profiles

PLOS ONE

Dear Dr. Jabbour,

Thank you for submitting your manuscript to PLOS ONE. After careful consideration, we feel that it has merit but does not fully meet PLOS ONE’s publication criteria as it currently stands. Therefore, we invite you to submit a revised version of the manuscript that addresses the points raised during the review process.

We would appreciate receiving your revised manuscript by Feb 01 2020 11:59PM. To enhance the reproducibility of your results, we recommend that if applicable you deposit your laboratory protocols in protocols.io, where a protocol can be assigned its own identifier (DOI) such that it can be cited independently in the future. For instructions see: http://journals.plos.org/plosone/s/submission-guidelines#loc-laboratory-protocols

We look forward to receiving your revised manuscript.

Kind regards,

Antonio Simone Laganà, M.D.

Academic Editor

PLOS ONE

Additional Editor Comments:

The topic of the manuscript is interesting. Nevertheless, the reviewers raised several concerns: considering this point, I invite authors to perform the required major revisions.

Journal Requirements:

Reviewers' comments:

Reviewer's Responses to Questions

**Comments to the Author**

1. Is the manuscript technically sound, and do the data support the conclusions?

Reviewer #1: Partly

Reviewer #2: Yes

Reviewer #3: Partly

2. Has the statistical analysis been performed appropriately and rigorously? 

Reviewer #1: Yes

Reviewer #2: No

Reviewer #3: No

3. Have the authors made all data underlying the findings in their manuscript fully available?

Reviewer #1: Yes

Reviewer #2: Yes

Reviewer #3: Yes

4. Is the manuscript presented in an intelligible fashion and written in standard English?

Reviewer #1: Yes

Reviewer #2: Yes

Reviewer #3: Yes

5. Review Comments to the Author

Reviewer #1: GENERAL COMMENTS

The manuscript entitled “Carotid intima-media thickness in polycystic ovary syndrome and its association with hormone and lipid profiles” refers to a topic on which there seems to be quite a consensus that CITM is greater in women with PCOS. In addition, numerous antecedents associate this parameter with cardiovascular risk and alterations of the lipid profile. It has also been pointed out a relationship between CIMT and the levels of androgens, one of the fundamental features of PCOS, so it is difficult for me to rescue the novelty of the article. I think it is essential that you can emphasize this aspect. Could it be that "oligomenorrheic patients revealed a relationship between CIMT and the suspected duration of disease" is a new finding?

SPECIFIC COMMENTS

Other articles have reported higher CIMT in young PCOS women, however most studies have failed to demonstrate greater cardiovascular risk in perimenopausal women with PCOS despite having a higher prevalence of metabolic disorders at an early age. It will be possible that when women approaching menopause, the control women worsen this parameter and the PCOS remain the same, equating CIMT at this stage? In other words, is it likely that the damage appears earlier, but not that there are major alterations in the vascular system? If there is background on this, I suggest adding and discussing.

Materials and methods (study participants), I would you like to know if they had a range of BMI for the recruitment of women (PCOS and controls), nothing is pointed out about it. The authors only mention that the control women had a BMI similar to PCOS.

In the selection of the control group, the authors mention that it was healthy and regularly menstruating female volunteers, it is striking that they did not rule out hyperandrogenism (at least clinical). In fact, table 1 shows that there is 18.6% of hirsutism and 14% of hyperandrogenemia in the control group. In addition, they do not mention whether the control group had ovarian ultrasound. Finally, it would be appropriate to be able to confirm that the percentage of control women with hyperandrogenism have ovaries without polycystic morphology. Otherwise the question remains whether they are pure hyperandrogenic or if they have the C phenotype of Rotterdam?

In this article, the authors diagnose PCOS according to the Rotterdam criteria, giving the frequency of each phenotype, however in the results they do not report this data associated with CIMT. What was the reason for reporting those frequencies by phenotype? On the other hand, this idea is retaken in the discussion (lines 292-295), but without mentioning the results of the present study.

The way that the results are organized in tables 1-3 is confusing. The data does not appear to be grouped in the proper order, to simplify reading and interpretation. I think the anthropometric, clinical and demographic data should go in the first table, since they are the ones obtained in the first instance; then hormonal, metabolic and CIMT measurement. The processed data (applying cut-off or classification values) should be shown in the final table or at the end of their respective tables (raw data). I suggest including the free androgen index, as it is a good reflection of hyperandrogenism (testosterone action).

In table 1, draws attention to the high percentage of acne, both in the PCOS and in the control group. Is there any bias in the selection of the groups? How was this parameter quantified (presence or absence)? How could these values be explained?

The most comparable results with this study would be those of early reproductive age women with PCOS, such as those suggested below and that have not been included in the discussion.

J Clin Endocrinol Metab. 2018;103(4):1622-1630. Meun C et al.

Menopause. 2012; 19(1):10–15. Munir JA et al.

Indian J Endocrinol Metab. 2016;20(5):662-666. Garg N et al.

Gynecol Endocrinol. 2015;31(6):477-82. Yilmaz SA et al.

Int J Prev Med. 2013;4(11):1266-70. Allameh Z et al.

Reviewer #2: Comments to the Author

This study aimed to investigate if CIMT is increased in PCOS patients compared to healthy controls and if there is an association with hormone and metabolic profiles.

This study presents interesting findings, but there are considerable concerns related to the study design and data presenting.

Specific comments are as follows:

1. Please explain the definition of biochemical hyperandrogenism. How is the local reference defined?

2. In abstract and method, age matched and similar BMI controls were enrolled. Please describe the criterion for matching in detail in the manuscript.

3. What is oligomenorrhea onset (years after menarche) as 0.0 (0.0-3.0) in Table 1?

Is “Suspected duration of disease since onset” not directly related to the age of the subjects?

4. Why did not the authors compare of CIMT between the control group and hyperandrogenic and non-hyperandrogenic PCOS patients?

5. The authors need to explain the table 5 in detail. They stated that multiple linear regression analysis was carried out in order to identify independent factors that predict CIMT as the dependent continuous variable. What is the dependent variable? If CIMT is the dependent variable, the authors should analyze using univariate linear regression analysis with CIMT as the dependent variable.

Reviewer #3: I was pleased to revise the manuscript entitled “Carotid intima-media thickness in polycystic ovary syndrome and its association with hormone and lipid profiles” (Manuscript Number: PONE-D-19-30548).

The study was approved by the Ethics Committee of the Medical University of Vienna (EK-Nr. 1197/2015). In general, this manuscript was aimed to investigate if the intima-media thickness of the common carotid artery is increased in PCOS patients compared to healthy controls and if there is an association with hormone and metabolic profiles, and in my opinion this study is interesting for the readers. Nevertheless, methodology is not accurate, and conclusions are not completely supported by the reported data. Authors should clarify some point and improve the results and discussion.

In general, the Manuscript may benefit from several major revisions, as suggested below:

1. Results and statistical methods. I would suggest investigating the multicollinearity between PCOS and cardiovascular risk factors. The strong association between them and PCOS may explain the cardiovascular risk reported in these patients. Age, BMI, and smoking status are not the only possible confounders in the association between PCOS and cardiovascular risk.

2. Methods. It is not clear why the Authors used the correlation coefficient instead of univariate linear regression.

3. Discussion. Lines 282. This point is unclear. The PCOS is a complex disorder and it is probably that specific included metabolic factors are the cause of increased CIMT in PCOS women. It is of paramount importance to identify these elements as possible target of preventive treatments.

4. Conclusion. Lines 391-393. Based in the results, the role of hyperandrogenism as crucial determinant of CIMT is not demonstrated.

5. Conclusion. Lines 394. This statement is not supported by results. A complete multivariate regression analysis was not performed, the collinearity needs to be better investigated and assessed. A backward method could be better with an appropriate evaluation of collinearity by the use of variance inflation factor.

6. I would suggest improving the introduction reporting about the role of insulin resistance, that is one of the most important mechanisms of PCOS pathogenesis. For this reason, the use of insulin-sensitizers, such an inositol isoform, gained increasing attention due to their safety profile and effectiveness. Authors may better discuss this point, taking to account these recent articles: PMID: 30270194.

6. PLOS authors have the option to publish the peer review history of their article (what does this mean?). If published, this will include your full peer review and any attached files.

Reviewer #1: No

Reviewer #2: No

Reviewer #3: No

---

## [Author Response · Author response to Decision Letter 0]

4 Feb 2020

Please find below our response to the points raised by the reviewers. The mentioned lines of the manuscript correspond to those of the document “Revised_Manuscript_With_Track_Changes_Rhea_Jabbour”, as they differ from the ones in the Manuscript without tracked changes.

Reviewer #1: 

GENERAL COMMENTS

The manuscript entitled “Carotid intima-media thickness in polycystic ovary syndrome and its association with hormone and lipid profiles” refers to a topic on which there seems to be quite a consensus that CITM is greater in women with PCOS. In addition, numerous antecedents associate this parameter with cardiovascular risk and alterations of the lipid profile. It has also been pointed out a relationship between CIMT and the levels of androgens, one of the fundamental features of PCOS, so it is difficult for me to rescue the novelty of the article. I think it is essential that you can emphasize this aspect. Could it be that "oligomenorrheic patients revealed a relationship between CIMT and the suspected duration of disease" is a new finding?

Reply: It is true that there is already a lot of data about CIMT and PCOS. However, discrepant findings have been reported regarding CIMT increase in PCOS, as described in our Introduction. The published results concerning predictors of CIMT are also still very controversial. Therefore, our aim was to verify those findings and see how it is in the Austrian population, given that there is no literature about it to our knowledge yet. Indeed, we found that CIMT is increased in PCOS and given the scope of PLOS ONE to contribute to the base of scientific knowledge by also publishing confirmatory studies, we were hoping that you would consider our manuscript for publication. 

On top of that, you are correct. To our knowledge, the fact that suspected duration of disease was found to be a predictor of CIMT in oligomenorrhoic patients is a new finding. We have therefore emphasized this aspect in the Discussion:

l.576-591: “Furthermore, our study revealed suspected duration of disease as predictor of CIMT, which is a new finding that has not been reported so far to our knowledge. Therefore, oligomenorrhoic patients seem to feature higher CIMT the earlier onset of disease occurs. Although suspected duration of disease is related to the subjects’ age, it also depends on the age of menarche that varies in each patient. Besides, PCOS patients have later menarche than healthy controls[30,64,65], therefore it does not totally equate to the age. The observed relationship between suspected duration of disease and CIMT in oligomenorrhoic patients thus suggests even more that PCOS itself contributes to the enhancement of atherosclerosis, probably due to hyperandrogenism, an adverse lipid profile and hyperinsulinemia.”

SPECIFIC COMMENTS

Other articles have reported higher CIMT in young PCOS women, however most studies have failed to demonstrate greater cardiovascular risk in perimenopausal women with PCOS despite having a higher prevalence of metabolic disorders at an early age. It will be possible that when women approaching menopause, the control women worsen this parameter and the PCOS remain the same, equating CIMT at this stage? In other words, is it likely that the damage appears earlier, but not that there are major alterations in the vascular system? If there is background on this, I suggest adding and discussing.

Reply: This is a very interesting point, thank you for your valuable input. We have found several studies and have added two of them in the Discussion l.592-599:

-Journal of Clinical Endocrinology and Metabolism. 2011. p. 3675–7. Fauser BCJM et al. [64]

-Climacteric. Taylor and Francis Ltd; 2017. p. 222–7. Gunning MN et al. [65]

We have also added your suggested reference: 

-J Clin Endocrinol Metab. 2018;103(4):1622-1630. Meun C et al. [30]

l.592-599: “However, although PCOS patients feature increased CIMT and metabolic and cardiovascular risk at a younger age, there is evidence suggesting that they do not show higher prevalence of cardiovascular events than controls after menopause [30,64,65]. Possible explanations could be a protective effect of hyperandrogenism, especially DHEAS, in menopause, or delayed menopause in PCOS[30,64,65]. Moreover, differences in cardiovascular risk factors (e.g. diabetes, abdominal obesity) between PCOS and controls seem to be less preponderant in aging women, explaining the similar cardiovascular morbidity and mortality later in life[65].” 

Materials and methods (study participants), I would you like to know if they had a range of BMI for the recruitment of women (PCOS and controls), nothing is pointed out about it. The authors only mention that the control women had a BMI similar to PCOS.

In the selection of the control group, the authors mention that it was healthy and regularly menstruating female volunteers, it is striking that they did not rule out hyperandrogenism (at least clinical). In fact, table 1 shows that there is 18.6% of hirsutism and 14% of hyperandrogenemia in the control group. In addition, they do not mention whether the control group had ovarian ultrasound. Finally, it would be appropriate to be able to confirm that the percentage of control women with hyperandrogenism have ovaries without polycystic morphology. Otherwise the question remains whether they are pure hyperandrogenic or if they have the C phenotype of Rotterdam?

Reply: Yes, we had a specific range of BMI for the recruitment of our study subjects. It is reported in the Materials and Methods as exclusion criteria in l.153, i.e. “BMI <17 kg/m2 or >50 kg/m2”.

Actually, we did rule out PCOS in controls with hyperandrogenism. In fact, 8 women of the control group showed hirsutism and 6 hyperandrogenemia. However, hyperandrogenemia was considered mandatory in order to fulfill the criterion of hyperandrogenism in this study (l.139-140). Therefore, in these 6 regularly-cycling women, we performed transvaginal ultrasound to make sure that they did not feature polycystic ovaries and did not belong to phenotype 3 of Rotterdam, as you correctly pointed out. Given that ultrasound was normal in these 6 patients, we then included them in our study. We did not mention this approach in the manuscript for the sake of space, but we have added it now in the Materials and Methods:

l.149-151: “PCOS was ruled out in all controls according to the Rotterdam ESHRE/ASRM criteria[3], as transvaginal ultrasound was performed in case of hyperandrogenism to exclude PCOS phenotype 3.”

l.170-171 “Pelvic ultrasound was further performed in PCOS patients and controls with hyperandrogenism. ”

In this article, the authors diagnose PCOS according to the Rotterdam criteria, giving the frequency of each phenotype, however in the results they do not report this data associated with CIMT. What was the reason for reporting those frequencies by phenotype? On the other hand, this idea is retaken in the discussion (lines 292-295), but without mentioning the results of the present study.

Reply: The frequency of each phenotype was only given in a descriptive manner, in order to characterize our study subjects in the most precise manner. As pointed out in the Discussion (l.621-623), the sample size was too small in order to be able to perform statistical analyses among each phenotype. Therefore, we did not mention the results of the present study concerning cardiovascular risk profile according to phenotypes in the Discussion either. 

The way that the results are organized in tables 1-3 is confusing. The data does not appear to be grouped in the proper order, to simplify reading and interpretation. I think the anthropometric, clinical and demographic data should go in the first table, since they are the ones obtained in the first instance; then hormonal, metabolic and CIMT measurement. The processed data (applying cut-off or classification values) should be shown in the final table or at the end of their respective tables (raw data). I suggest including the free androgen index, as it is a good reflection of hyperandrogenism (testosterone action).

In table 1, draws attention to the high percentage of acne, both in the PCOS and in the control group. Is there any bias in the selection of the groups? How was this parameter quantified (presence or absence)? How could these values be explained?

Reply: Thank you for your comment. In fact, given that we have a lot of data, we tried to dispose it as clearly as possible in as few tables as possible. As you suggested, we have added anthropometric data, i.e. WC, HC and WHR, in Table 1 and TC/HDL > 4 in Table 3. However, in our opinion, merging hormonal, metabolic and CIMT measurements in one table would lead to a table that is too long and unclear, so we would prefer to keep two tables, i.e. Table 2 and 3. Moreover, cut-off values for hyperandrogenism were not added in Table 3, as this would make the table too long as well. Besides, there are part of the baseline classification of the study population and should therefore be presented in Table 1 in our opinion. Perhaps the added subclassification with “*” makes it more clear. We have therefore modified the following sentence and tables:

l.251-252: “Baseline demographic and anthropometrical parameters and clinical characteristics of the study population are exposed in Table 1.”

l.256-260: “PCOS patients also exhibited preponderance of abdominal obesity, as revealed by significantly increased WC, WHR (P<0.001) and HC (P=0.019) and elevated prevalence of WC>80 cm (P=0.001). As expected[3], they further featured significantly higher prevalence of ovulatory dysfunction, hirsutism and hyperandrogenemia (P<0.001) and increased age at menarche (P=0.010).”

Regarding the free androgen index, thank you for your suggestion, we included it into our analyses now. It is exposed in Tables 3 and 4, as well as in the text as follows:

l.175-176: “Free androgen index (FAI) was calculated as follows: (total testosterone[nmol/L]/SHBG[nmol/L])x100[32].”

l.390-394: “PCOS patients further presented with significantly higher levels of all measured androgens, FAI, 17-OHP, luteinizing hormone (LH), LH/FSH (follicle-stimulating hormone) ratio and Anti-Müllerian hormone (AMH) (P≤0.001), and lower levels of SHBG (P=0.006) and FSH (P=0.008).” 

l.408-412: “Several cardiovascular risk factors and hormonal parameters were found to be significantly positively correlated with CIMT in our subjects (Table 4), including total testosterone, free testosterone, androstenedione, FAI, AMH, LH/FSH ratio, WC, WHR, BMI, FGS and apolipoprotein B (P<0.001), as well as 17-OHP, LDL-C, smoking, TC, triglycerides, TC/HDL-C ratio, DHEAS and SBP (P<0.05).”

l.540-558: “Besides, the role of testosterone as crucial determinant of CIMT increase is further emphasized by the positive relationship of CIMT seen with FAI, which represents a good reflection of testosterone action.”

Concerning acne, its presence was quantified subjectively, due to patient reports or when clinically visible. In fact, the high prevalence could be explained by the fact that we included a lot of young subjects in our study. On top of that, acne is not a reliable reflection of hyperandrogenemia, therefore hyperandrogenemia was considered mandatory in order to fulfill the criterion of hyperandrogenism in this study, as explained in l. 139-140. We have added the following sentence in the Discussion:

l.609-617: “Besides, the presence of acne was quantified subjectively, thus no major focus should be put on its prevalence.” 

The most comparable results with this study would be those of early reproductive age women with PCOS, such as those suggested below and that have not been included in the discussion.

J Clin Endocrinol Metab. 2018;103(4):1622-1630. Meun C et al. 

Menopause. 2012; 19(1):10–15. Munir JA et al.

Indian J Endocrinol Metab. 2016;20(5):662-666. Garg N et al.

Gynecol Endocrinol. 2015;31(6):477-82. Yilmaz SA et al.

Int J Prev Med. 2013;4(11):1266-70. Allameh Z et al.

Reply: Thank you very much for your suggestion, we have already included a lot of references in our manuscript, so we cannot cite all of them. We had a look at the five papers you have mentioned and decided to include the following ones: 

-J Clin Endocrinol Metab. 2018;103(4):1622-1630. Meun C et al. [30]

l. 89-91: In fact, hyperandrogenism is said to lead to an increase in CIMT through its proatherogenic effect[20], but numerous studies have also demonstrated an inverse correlation of CIMT with androgen levels in women[27,28,29,30].

l. 576-588: “Furthermore, our study revealed suspected duration of disease as predictor of CIMT, which is a new finding that has not been reported so far to our knowledge. Therefore, oligomenorrhoic patients seem to feature higher CIMT the earlier onset of disease occurs. Although suspected duration of disease is related to the subjects’ age, it also depends on the age of menarche that varies in each patient. Besides, PCOS patients have later menarche than healthy controls[30,64,65], therefore it does not totally equate to the age.” 

l.592-596: “However, although PCOS patients feature increased CIMT and metabolic and cardiovascular risk at a younger age, there is evidence suggesting that they do not show higher prevalence of cardiovascular events than controls after menopause [30,64,65]. Possible explanations could be a protective effect of hyperandrogenism, especially DHEAS, in menopause, or delayed menopause in PCOS[30,64,65]. 

-Indian J Endocrinol Metab. 2016;20(5):662-666. Garg N et al. [34] 

l.449-451:“There is evidence that women with PCOS feature an increased CIMT compared to healthy controls[20,21,34], with reported mean differences in CIMT ranging from 0.06 mm[35] to 0.14 mm[24,36].”

However, we did not include the following ones and hope that this is okay for the reviewer: 

-Menopause. 2012; 19(1):10–15. Munir JA et al.: In fact, Munir et al. examined perimenopausal women, with a mean age of 50 years, whereas we had women of reproductive age between 18 and 34 years.

-Gynecol Endocrinol. 2015;31(6):477-82. Yilmaz SA et al.: In fact, authors examined radial IMT and not carotid IMT.

-Int J Prev Med. 2013;4(11):1266-70. Allameh Z et al.: In fact, Allameh et al. examined women between 35 and 50 years, with a mean age of 38 years, which might also include perimenopausal women, whereas we had an age-range of 18 to 34 years.

If the reviewer still wants us to include these references, we shall be happy to do so in a future revision process.

 

Reviewer #2: 

1. Please explain the definition of biochemical hyperandrogenism. How is the local reference defined?

Reply: The definition of biochemical hyperandrogenism is exposed in the Materials and Methods l.136-139. We have added the word “normal” in l.137 to make it more clear.

l.136-139: “Biochemical hyperandrogenism was defined according to the local laboratory’s normal reference ranges as at least one of the following conditions: dehydroepiandrosterone sulfate (DHEAS) >3.7 μg/mL, free testosterone >0.22 ng/mL, total testosterone >0.48 ng/mL, androstenedione >4.1 ng/mL.”

2. In abstract and method, age matched and similar BMI controls were enrolled. Please describe the criterion for matching in detail in the manuscript.

Reply: Thank you for pointing that out. We have added this information in the Materials and Methods l.215-223:

l.215-223:“Age matching was achieved by ensuring that overall age distribution in terms of mean and standard deviation in both groups is roughly the same.”

3. What is oligomenorrhea onset (years after menarche) as 0.0 (0.0-3.0) in Table 1?

Is “Suspected duration of disease since onset” not directly related to the age of the subjects?

Reply: We have added the following explanation in the Materials and Methods l.164-166: 

l.164-166: “We further defined the suspected starting point of disease in PCOS patients as the onset of oligomenorrhea. Therefore, when oligomenorrhea started right at menarche, it was noted as starting 0 years after menarche, meaning that 0 years equals menarche.” 

Concerning the relation of “suspected duration of disease” and “age”, we have added the following explanation in the Discussion l.576-588:

l.576-588: “Furthermore, our study revealed suspected duration of disease as predictor of CIMT, which is a new finding that has not been reported so far to our knowledge. Therefore, oligomenorrhoic patients seem to feature higher CIMT the earlier onset of disease occurs. Although suspected duration of disease is related to the subjects’ age, it also depends on the age of menarche that varies in each patient. Besides, PCOS patients have later menarche than healthy controls[30,64,65], therefore it does not totally equate to the age.” 

4. Why did not the authors compare of CIMT between the control group and hyperandrogenic and non-hyperandrogenic PCOS patients?

Reply: Our sample size was too small to be able to compare CIMT between controls and hyperandrogenic and non-hyperandrogenic PCOS patients. A larger sample size would be needed in order to yield statistical significance, as described in the Discussion l.603-609 and l.621-623.

5. The authors need to explain the table 5 in detail. They stated that multiple linear regression analysis was carried out in order to identify independent factors that predict CIMT as the dependent continuous variable. What is the dependent variable? If CIMT is the dependent variable, the authors should analyze using univariate linear regression analysis with CIMT as the dependent variable.

Reply: In fact, as explained in the Results l.419-420, as well as in the title of Table 5, CIMT is the dependent variable. As stated in l.420-422, model 1 revealed that the diagnosis of PCOS was the strongest predictor of CIMT when considering PCOS patients and controls as a group, explaining 70% of its variability, as quantified by the coefficient of determination R2, which expresses the proportion of variability in the dependent variable, i.e. CIMT, explained by the model (see definition l.238-240). In order to show that PCOS remains the primary predictor of CIMT, independent of BMI, age and smoking status, multiple adjustments had to be made for these 3 variables (l.422-424). This required a multiple linear regression analysis and not univariate linear regression analysis. 

 

Reviewer #3:

Nevertheless, methodology is not accurate, and conclusions are not completely supported by the reported data. Authors should clarify some point and improve the results and discussion.

Reply: Thank you for your feedback. We have improved the Results and Discussion by modifying Tables 1-4 (see above), including the free androgen index (FAI) into our analyses and adding the following elements:

Results:

l.251-252: “Baseline demographic and anthropometrical parameters and clinical characteristics of the study population are exposed in Table 1.”

l.256-260: “PCOS patients also exhibited preponderance of abdominal obesity, as revealed by significantly increased WC, WHR (P<0.001) and HC (P=0.019) and elevated prevalence of WC>80 cm (P=0.001). As expected[3], they further featured significantly higher prevalence of ovulatory dysfunction, hirsutism and hyperandrogenemia (P<0.001) and increased age at menarche (P=0.010).”

l.390-394: “PCOS patients further presented with significantly higher levels of all measured androgens, FAI, 17-OHP, luteinizing hormone (LH), LH/FSH (follicle-stimulating hormone) ratio and Anti-Müllerian hormone (AMH) (P≤0.001), and lower levels of SHBG (P=0.006) and FSH (P=0.008).” 

l.408-412: “Several cardiovascular risk factors and hormonal parameters were found to be significantly positively correlated with CIMT in our subjects (Table 4), including total testosterone, free testosterone, androstenedione, FAI, AMH, LH/FSH ratio, WC, WHR, BMI, FGS and apolipoprotein B (P<0.001), as well as 17-OHP, LDL-C, smoking, TC, triglycerides, TC/HDL-C ratio, DHEAS and SBP (P<0.05).”

Discussion:

l.540-558: “Besides, the role of testosterone as crucial determinant of CIMT increase is further emphasized by the positive relationship of CIMT seen with FAI, which represents a good reflection of testosterone action.”

l.576-591: “Furthermore, our study revealed suspected duration of disease as predictor of CIMT, which is a new finding that has not been reported so far to our knowledge. Therefore, oligomenorrhoic patients seem to feature higher CIMT the earlier onset of disease occurs. Although suspected duration of disease is related to the subjects’ age, it also depends on the age of menarche that varies in each patient. Besides, PCOS patients have later menarche than healthy controls[30,64,65], therefore it does not totally equate to the age. The observed relationship between suspected duration of disease and CIMT in oligomenorrhoic patients thus suggests even more that PCOS itself contributes to the enhancement of atherosclerosis, probably due to hyperandrogenism, an adverse lipid profile and hyperinsulinemia.”

l.592-599: “However, although PCOS patients feature increased CIMT and metabolic and cardiovascular risk at a younger age, there is evidence suggesting that they do not show higher prevalence of cardiovascular events than controls after menopause [30,64,65]. Possible explanations could be a protective effect of hyperandrogenism, especially DHEAS, in menopause, or delayed menopause in PCOS[30,64,65]. Moreover, differences in cardiovascular risk factors (e.g. diabetes, abdominal obesity) between PCOS and controls seem to be less preponderant in aging women, explaining the similar cardiovascular morbidity and mortality later in life[65].” 

l.609-617: “Besides, the presence of acne was quantified subjectively, thus no major focus should be put on its prevalence.” 

1. Results and statistical methods. I would suggest investigating the multicollinearity between PCOS and cardiovascular risk factors. The strong association between them and PCOS may explain the cardiovascular risk reported in these patients. Age, BMI, and smoking status are not the only possible confounders in the association between PCOS and cardiovascular risk.

Reply: Our primary hypothesis was to demonstrate that there is a difference in CIMT between PCOS and controls. We succeeded in showing this in our study. However, in order to analyze the exact factors that play a role in CIMT increase in PCOS, our sample size was too small. Therefore, we limited ourselves to a model showing that even when taking the most known confounders into account, i.e. age, BMI and smoking status, PCOS was still the primary predictor of CIMT. 

However, we did explore multicollinearity between PCOS and cardiovascular risk factors. In fact, as described in the Materials and Methods l.230-241, after adjusting for BMI, age and smoking status, factors found to be significantly associated with CIMT in individual bivariate correlation were entered sequentially as independent factors into the regression model. First, the independent variable best correlated with CIMT was included, then the one with the next highest correlation, checking for multicollinearity and normal distribution of the residuals. However, no statistical significance was found for further hormonal and cardiovascular risk factors, i.e. androgens or lipidemic parameters, probably due to multicollinearity with the PCOS status (l.430-432). They were therefore excluded from the predictive model.

Due to high multicollinearity and the small sample size of the study, we thus decided not to publish these results or investigate further, given that the small sample size could lead to a bias. We have therefore only created a model with age, BMI and smoking status in order to subtract their potential confounding effect on CIMT (see Table 5). As pointed out in the Discussion (l.621-623), a larger sample size is needed in order to investigate the effect of other cardiovascular risk factors found to be significantly associated with CIMT in correlation analysis, like hyperandrogenism for example. 

2. Methods. It is not clear why the Authors used the correlation coefficient instead of univariate linear regression.

Reply: Univariate linear regression analysis would not add any new information to the one obtained with Spearman’s rank correlation analysis (rho). That is why solely correlation and multiple linear regression analyses were performed in this study. Besides, unlike univariate linear regression analysis, Spearman’s rank coefficient does not assume linear dependencies and it is also more robust to outliers. Therefore, it is in our opinion a more adequate statistical tool for analysis of our data. 

3. Discussion. Lines 282. This point is unclear. The PCOS is a complex disorder and it is probably that specific included metabolic factors are the cause of increased CIMT in PCOS women. It is of paramount importance to identify these elements as possible target of preventive treatments.

Reply: The primary aim of our study was to demonstrate that there is a difference in CIMT between PCOS and controls. We succeeded in showing this in our research. In order to analyze the exact factors that play a role in CIMT increase in PCOS, our sample size was too small. However, with our results from correlation and multiple regression analysis, we tried to identify those elements in an explorative manner. There are exposed in the Discussion l. 460-591. In fact, the sentence “These findings suggest that the disorder itself is playing a causative role in CIMT increase” was only supposed to serve as an introduction for the following detailed analysis of those parameters.

4. Conclusion. Lines 391-393. Based in the results, the role of hyperandrogenism as crucial determinant of CIMT is not demonstrated.

Reply: It is true that we did not include hyperandrogenemia into our regression model, due to multicollinearity with the PCOS status. However, in correlation analysis, markers of hyperandrogenemia, i.e. total testosterone, free testosterone and androstenedione showed the strongest positive correlation with CIMT (P<0.001) among all analyzed cardiovascular risk factors. Besides, unpaired two-tailed Student’s t-test revealed significantly higher CIMT values among the entire study population in case of hyperandrogenemia (0.47 ± 0.06 (n=38) vs. 0.40 ± 0.07 (n=46), P<0.001). Therefore, the role of hyperandrogenism as crucial determinant of CIMT is demonstrated in our opinion. In order to emphasize our results, we have now included the free androgen index (FAI) into our analyses, given that it represents a good reflection of testosterone action. In fact, FAI was significantly higher in PCOS patients (P<0.001, see Table 3) and it was also significantly positively correlated with CIMT (P<0.001, see Table 4). We have therefore added the following information into our manuscript:

l.175-176: “Free androgen index (FAI) was calculated as follows: (total testosterone[nmol/L]/SHBG[nmol/L])x100[32].”

l.390-394: “PCOS patients further presented with significantly higher levels of all measured androgens, FAI, 17-OHP, luteinizing hormone (LH), LH/FSH (follicle-stimulating hormone) ratio and Anti-Müllerian hormone (AMH) (P≤0.001), and lower levels of SHBG (P=0.006) and FSH (P=0.008).” 

l.408-412: “Several cardiovascular risk factors and hormonal parameters were found to be significantly positively correlated with CIMT in our subjects (Table 4), including total testosterone, free testosterone, androstenedione, FAI, AMH, LH/FSH ratio, WC, WHR, BMI, FGS and apolipoprotein B (P<0.001), as well as 17-OHP, LDL-C, smoking, TC, triglycerides, TC/HDL-C ratio, DHEAS and SBP (P<0.05).”

l.540-558: “Besides, the role of testosterone as crucial determinant of CIMT increase is further emphasized by the positive relationship of CIMT seen with FAI, which represents a good reflection of testosterone action.”

5. Conclusion. Lines 394. This statement is not supported by results. A complete multivariate regression analysis was not performed, the collinearity needs to be better investigated and assessed. A backward method could be better with an appropriate evaluation of collinearity by the use of variance inflation factor.

Reply: We would kindly refer the reviewer to our reply to comment 1, as we can give the same response to this comment as well. 

6. I would suggest improving the introduction reporting about the role of insulin resistance, that is one of the most important mechanisms of PCOS pathogenesis. For this reason, the use of insulin-sensitizers, such an inositol isoform, gained increasing attention due to their safety profile and effectiveness. Authors may better discuss this point, taking to account these recent articles: PMID: 30270194.

Reply: Thank you for your valuable comment, we have added the suggested reference in the Introduction l.60-62: 

l.60-62: “IR being a key component of the pathophysiology of PCOS, treatment options with insulin-sensitizers, such as metformin or inositol isoforms[11], are increasingly being used due to their beneficial effects on metabolic and hormonal parameters.”

---

## [Decision Letter · Decision Letter 1]

13 Mar 2020

PONE-D-19-30548R1

Carotid intima-media thickness in polycystic ovary syndrome and its association with hormone and lipid profiles

PLOS ONE

Dear Dr. Jabbour,

Thank you for submitting your manuscript to PLOS ONE. After careful consideration, we feel that it has merit but does not fully meet PLOS ONE’s publication criteria as it currently stands. Therefore, we invite you to submit a revised version of the manuscript that addresses the points raised during the review process.

We would appreciate receiving your revised manuscript by Apr 27 2020 11:59PM. To enhance the reproducibility of your results, we recommend that if applicable you deposit your laboratory protocols in protocols.io, where a protocol can be assigned its own identifier (DOI) such that it can be cited independently in the future. For instructions see: http://journals.plos.org/plosone/s/submission-guidelines#loc-laboratory-protocols

We look forward to receiving your revised manuscript.

Kind regards,

Antonio Simone Laganà, M.D., Ph.D.

Academic Editor

PLOS ONE

Additional Editor Comments (if provided):

Authors performed the required changes, which were positively evaluated by the reviewers, and improved the quality of the manuscript.

Nevertheless, some of them asked for other additional minor revisions: for this reason, I invite authors to perform these additional changes.

Reviewers' comments:

Reviewer's Responses to Questions

**Comments to the Author**

1. If the authors have adequately addressed your comments raised in a previous round of review and you feel that this manuscript is now acceptable for publication, you may indicate that here to bypass the “Comments to the Author” section, enter your conflict of interest statement in the “Confidential to Editor” section, and submit your "Accept" recommendation.

Reviewer #1: (No Response)

Reviewer #2: All comments have been addressed

Reviewer #3: All comments have been addressed

2. Is the manuscript technically sound, and do the data support the conclusions?

Reviewer #1: Partly

Reviewer #2: Yes

Reviewer #3: Yes

3. Has the statistical analysis been performed appropriately and rigorously? 

Reviewer #1: Yes

Reviewer #2: Yes

Reviewer #3: Yes

4. Have the authors made all data underlying the findings in their manuscript fully available?

Reviewer #1: Yes

Reviewer #2: Yes

Reviewer #3: Yes

5. Is the manuscript presented in an intelligible fashion and written in standard English?

Reviewer #1: Yes

Reviewer #2: Yes

Reviewer #3: Yes

6. Review Comments to the Author

Reviewer #1: The authors have addressed all the comments of the reviewers and the manuscript has been improved. However, I have additional minor comments:

-In the sentence (592-599): “Possible explanations could be a protective effect of hyperandrogenism, especially DHEAS, in menopause, or delayed menopause in PCOS.

I disagree with this statement and I find it contradictory to what is stated in the article. It seems to me that it is unlikely that hyperandrogenism (DHEAS) has a protective effect. Probably to be an estrogenic effect, as some articles show. “Meun et al. hypothesize that later menopause with prolonged estrogen exposure may have a protective role against CVD. (Meun, C., J Clin Endocrinol Metab. 2018; 103: 1622–1630).

Although some articles have reported that PCOS women have lower basal estradiol level, it has also been described that this value would be comparable to control women. Moreover, a study has showed higher peak estradiol level in PCOS patients (Eur J Obstet Gynecol Reprod Biol. 2013 Sep;170(1):165-70). Which could favor the testosterone/estradiol ratio towards estradiol effect as women with PCOS are older and androgens decreased.

-The range of BMI that the authors report is quite wide (17-50 kg/m2). Atypical outcomes are usually observed in women with BMI> 40. Did you observe any particularity in them or behave within the average? Remember the contribution of steroids in the adipose tissue of women with high BMI.

- Regarding transvaginal ultrasound that was performed in 6 regularly-cycling women in order to exclude PCOS phenotype. How long after the initial study was performed? Remember that the ovarian image may vary over time. I suggest indicating in materials and methods so that the interpretation is at the reader criteria

Reviewer #2: The authors revised the manuscript generally well according to my comments.

However, I still have a question. Have the authors analyzed if there is a difference in CIMT between obese PCOS patients and non-obese PCOS patients? It is recommended to present the results of this subgroup analysis.

Reviewer #3: The Authors clarified different points of the analysis improving the overall value of the manuscript that can be considered for publication.

7. PLOS authors have the option to publish the peer review history of their article (what does this mean?). If published, this will include your full peer review and any attached files.

Reviewer #1: No

Reviewer #2: No

Reviewer #3: No

---

## [Author Response · Author response to Decision Letter 1]

10 Apr 2020

Please find below our response to the points raised by the reviewers. The mentioned lines of the manuscript correspond to those of the document “Revised_Manuscript_With_Track_Changes_Rhea_Jabbour”, as they differ from the ones in the Manuscript without tracked changes.

Reviewer #1:

The authors have addressed all the comments of the reviewers and the manuscript has been improved. However, I have additional minor comments:

-In the sentence (592-599): “Possible explanations could be a protective effect of hyperandrogenism, especially DHEAS, in menopause, or delayed menopause in PCOS.” I disagree with this statement and I find it contradictory to what is stated in the article. It seems to me that it is unlikely that hyperandrogenism (DHEAS) has a protective effect. Probably to be an estrogenic effect, as some articles show. “Meun et al. hypothesize that later menopause with prolonged estrogen exposure may have a protective role against CVD. (Meun, C., J Clin Endocrinol Metab. 2018; 103: 1622–1630). 

Although some articles have reported that PCOS women have lower basal estradiol level, it has also been described that this value would be comparable to control women. Moreover, a study has showed higher peak estradiol level in PCOS patients (Eur J Obstet Gynecol Reprod Biol. 2013 Sep;170(1):165-70). Which could favor the testosterone/estradiol ratio towards estradiol effect as women with PCOS are older and androgens decreased.

Reply: Thank you very much for your valuable comment. We have clarified this point as exposed below.

l.388-396: “However, although women with PCOS feature increased CIMT and metabolic and cardiovascular risk at a younger age, there is evidence suggesting that they do not show higher prevalence of cardiovascular events than controls after menopause[30,64,65]. Possible explanations could be a protective effect of delayed menopause with a consequently prolonged estrogen exposure in PCOS[30,64,65] or even hyperandrogenism itself[64,65] in peri- and postmenopausal PCOS patients, as suggested by previous studies[27,28,29], probably mainly due to enzymatic conversion to estrogen. Moreover, differences in cardiovascular risk factors (e.g. diabetes, abdominal obesity) between PCOS and controls seem to be less preponderant in aging women, explaining the similar cardiovascular morbidity and mortality later in life[65].”

-The range of BMI that the authors report is quite wide (17-50 kg/m2). Atypical outcomes are usually observed in women with BMI> 40. Did you observe any particularity in them or behave within the average? Remember the contribution of steroids in the adipose tissue of women with high BMI.

Reply: We have only included 4 PCOS patients and no controls with a BMI > 40 kg/m2 in our study. Therefore, the sample size was too small in order to be able to analyze any atypical outcomes in these subjects.

-Regarding transvaginal ultrasound that was performed in 6 regularly-cycling women in order to exclude PCOS phenotype. How long after the initial study was performed? Remember that the ovarian image may vary over time. I suggest indicating in materials and methods so that the interpretation is at the reader criteria

Reply: Thank you very much for your suggestion. In fact, transvaginal ultrasound was performed either on the same day of blood tests, if past medical history suggested hyperandrogenemia, or the following day, once laboratory results were obtained. It was therefore performed between the third and the fifth day of the subjects’ menstrual cycle. We have included this precision in the Materials and Methods l.121-123:

l.121-123: “PCOS was ruled out in all controls according to the Rotterdam ESHRE/ASRM criteria[3], as transvaginal ultrasound was performed between the third and the fifth day of their menstrual cycle in case of hyperandrogenism to exclude PCOS phenotype 3.”

 

Reviewer #2: 

The authors revised the manuscript generally well according to my comments.

However, I still have a question. Have the authors analyzed if there is a difference in CIMT between obese PCOS patients and non-obese PCOS patients? It is recommended to present the results of this subgroup analysis.

Reply: Thank you very much for your feedback. Our sample size was too small in order to be able to compare CIMT between obese and non-obese PCOS patients. A larger sample size would be needed in order to yield statistical significance, as described in the Discussion l.418-420.

However, the impact of obesity on CIMT was analyzed in this study among the entire study population. Our results showed that visceral obesity was positively associated with CIMT, both in correlation analysis (see l.258-259) and multiple linear regression analysis (see l.274-275), as BMI represented an independent positive predictor of CIMT according to model 2. However, as stated in l.276-278, “PCOS status remained the primary predictor of CIMT, even after multiple adjustments for BMI, age and smoking status (P<0.001, R2=0.73, models 3 and 4), therefore indicating an independent effect of PCOS on CIMT (β=0.797, P<0.001)”.

Reviewer #3:

The Authors clarified different points of the analysis improving the overall value of the manuscript that can be considered for publication.

Reply: Thank you very much for your feedback and your positive recommendation.

---

## [Editor Report · Decision Letter 2]

13 Apr 2020

Carotid intima-media thickness in polycystic ovary syndrome and its association with hormone and lipid profiles

PONE-D-19-30548R2

Dear Dr. Jabbour,

We are pleased to inform you that your manuscript has been judged scientifically suitable for publication and will be formally accepted for publication once it complies with all outstanding technical requirements.

With kind regards,

Antonio Simone Laganà, M.D., Ph.D.

Academic Editor

PLOS ONE

Additional Editor Comments (optional):

Authors performed the required corrections. I am pleased to accept this paper for publication.
---

## [Editor Report · Acceptance letter]

15 Apr 2020

PONE-D-19-30548R2 

Carotid intima-media thickness in polycystic ovary syndrome and its association with hormone and lipid profiles 

Dear Dr. Jabbour:

I am pleased to inform you that your manuscript has been deemed suitable for publication in PLOS ONE. Congratulations! Your manuscript is now with our production department. 

With kind regards,

on behalf of

Dr. Antonio Simone Laganà 

Academic Editor

PLOS ONE